# Association of Corticosteroid Inhaler Type with Saliva Microbiome in Moderate-to-Severe Pediatric Asthma

**DOI:** 10.3390/biomedicines13010089

**Published:** 2025-01-02

**Authors:** Amir Hossein Alizadeh Bahmani, Mahmoud I. Abdel-Aziz, Simone Hashimoto, Corinna Bang, Susanne Brandstetter, Paula Corcuera-Elosegui, Andre Franke, Mario Gorenjak, Susanne Harner, Parastoo Kheiroddin, Leyre López-Fernández, Anne H. Neerincx, Maria Pino-Yanes, Uroš Potočnik, Olaia Sardón-Prado, Antoaneta A. Toncheva, Christine Wolff, Michael Kabesch, Aletta D. Kraneveld, Susanne J. H. Vijverberg, Anke H. Maitland-van der Zee

**Affiliations:** 1Department of Pulmonary Medicine, Amsterdam UMC Location University of Amsterdam, Meibergdreef 9, 1105 AZ Amsterdam, The Netherlands; 2Amsterdam Institute for Infection and Immunity, Inflammatory Diseases, 1105 AZ Amsterdam, The Netherlands; 3Amsterdam Public Health, Personalized Medicine, 1105 AZ Amsterdam, The Netherlands; 4Department of Pediatric Pulmonology and Allergy, Emma Children’s Hospital, Amsterdam University Medical Center, 1105 AZ Amsterdam, The Netherlands; 5Institute of Clinical Molecular Biology, Christian-Albrechts-University of Kiel, D-24105 Kiel, Germany; 6University Children’s Hospital Regensburg (KUNO), University of Regensburg, D-93049 Regensburg, Germany; 7Division of Pediatric Respiratory Medicine, Hospital Universitario Donostia, 20014 San Sebastián, Spain; 8Center for Human Molecular Genetics and Pharmacogenomics, Faculty of Medicine, University of Maribor, 2000 Maribor, Slovenia; 9Department of Pediatric Pneumology and Allergy, University Children’s Hospital Regensburg (KUNO), D-93049 Regensburg, Germany; 10Genomics and Health Group, Department of Biochemistry, Microbiology, Cell Biology and Genetics, Universidad de La Laguna (ULL), 38200 Santa Cruz de Tenerife, Spain; 11CIBER de Enfermedades Respiratorias, Instituto de Salud Carlos III, 28029 Madrid, Spain; 12Instituto de Tecnologías Biomédicas (ITB), Universidad de La Laguna (ULL), 38200 La Laguna, Spain; 13Department of Pediatrics, School of Medicine and Nursery, University of te Basque Country, 20014 San Sebastián, Spain; 14Division of Pharmacology, Utrecht Institute for Pharmaceutical Sciences, Faculty of Science, Utrecht University, Universiteitsweg 99, 3584 CG Utrecht, The Netherlands

**Keywords:** asthma, inhaled corticosteroids, metered-dose inhaler, dry powder inhaler, microbiome, saliva

## Abstract

**Background/Objectives**: Metered-dose inhalers (MDIs) and dry powder inhalers (DPIs) are common inhaled corticosteroid (ICS) inhaler devices. The difference in formulation and administration technique of these devices may influence oral cavity microbiota composition. We aimed to compare the saliva microbiome in children with moderate-to-severe asthma using ICS via MDIs versus DPIs. **Methods**: Saliva samples collected from 143 children (6–17 yrs) with moderate-to-severe asthma across four European countries (The Netherlands, Germany, Spain, and Slovenia) as part of the SysPharmPediA cohort were subjected to 16S rRNA sequencing. The microbiome was compared using global diversity (α and β) between two groups of participants based on inhaler devices (MDI (n = 77) and DPI (n = 65)), and differential abundance was compared using the Analysis of Compositions of Microbiomes with the Bias Correction (ANCOM-BC) method. **Results**: No significant difference was observed in α-diversity between the two groups. However, β-diversity analysis revealed significant differences between groups using both Bray–Curtis and weighted UniFrac methods (adjusted *p*-value = 0.015 and 0.044, respectively). Significant differential abundance between groups, with higher relative abundance in the MDI group compared to the DPI group, was detected at the family level [Carnobacteriaceae (adjusted *p* = 0.033)] and at the genus level [*Granulicatella* (adjusted *p* = 0.021) and *Aggregatibacter* (adjusted *p* = 0.011)]. **Conclusions**: Types of ICS devices are associated with different saliva microbiome compositions in moderate-to-severe pediatric asthma. The causal relation between inhaler types and changes in saliva microbiota composition needs to be further evaluated, as well as whether this leads to different potential adverse effects in terms of occurrence and level of severity.

## 1. Introduction

Asthma is a chronic inflammatory airway disease that influences millions of children worldwide. Inhaled corticosteroids (ICS) are among the main choices in managing and treating asthma and are recommended for mild-to-severe asthma in children and adults [1,2]. While ICS is highly efficient in controlling symptoms and reducing exacerbations [2], there is growing interest in understanding the potential impacts of ICS on the airway microbiome. A recent longitudinal study on the asthma population showed that ICS use can alter airway microbiome, including both fungal and bacterial microbiota [3]. Upper-airway microbiome (nasal and saliva) has been reported to be associated with asthma exacerbation among asthma patients on ICS therapy [4].

The type of inhalation devices that deliver ICS may play a role in shaping the oral and airway microbiome due to the differences in particle size, administration procedures, deposition patterns, and formulations. Metered-dose inhalers (MDIs) and dry powder inhalers (DPIs) are common ICS inhaler devices [5,6]. The risk of oral candidiasis and dysphonia was higher by using MDI devices than DPI devices when compared to a placebo group [7]. MDI inhalers are well-known, compact, convenient devices requiring actuation coordination to be used correctly. The breath and actuation coordination sometimes makes using the MDI inhalers difficult for some patients with low cognitive ability or difficulty actuating. MDI can be used with a spacer to improve medication delivery by extending the time of inhaling and allowing the lungs to absorb the medication more slowly and smoothly. In addition, using a spacer simplifies the inhaling process by reducing the need for actuation coordination in MDI users, especially in children [8]. MDI inhalers are cheaper than DPIs, and propellants are essential for their formulation. Breath–dose coordination is unnecessary while using DPI inhalers because DPIs are designed to release the medication in response to the patient’s inhalation effort. However, sufficient respiratory force is required for DPI inhalers to inhale the powder effectively. User instruction is inconsistent among different DPI inhalers, and a preparation step is usually needed before inhalation [5,6,9,10].

The lung microbiome is influenced by the upper respiratory tract and oropharynx microbiome [11]. Saliva offers a non-invasive sampling method for microbiome studies, which is particularly beneficial for children [12]. It has been previously shown that ICS treatment is associated with airway microbiome composition and diversity [3,13]. However, more research is needed to understand the association between ICS device types and salivary microbiome in children to optimize risk assessment, monitoring, and management strategies to protect patients’ oral and overall health under ICS treatments.

We hypothesized that differences in formulation and ICS device types may influence oral cavity microbiota composition. This study aims to compare the saliva bacterial microbiome in children with moderate-to-severe asthma under regular ICS treatment via MDI versus DPI devices.

## 2. Materials and Methods

### 2.1. Study Design

The Systems Pharmacology Approach to Uncontrolled Pediatric Asthma (SysPharmPediA) study is a multicenter European observational study with a case–control setting. The study design and population were described previously in detail by Abdel-Aziz et al. [14]. Briefly, children aged 6–17 with doctor-diagnosed moderate-to-severe asthma were included at tertiary hospitals from four European countries: The Netherlands, Germany, Spain, and Slovenia. This study with registration identifier NCT04865575 (at www.clinicaltrials.gov accessed on 29 April 2021) was carried out based on the Helsinki Declaration and approved by the medical ethics committee of the University Medical Center Utrecht, The Netherlands (NL55788.041.15); the National Medical Ethics Committee, Slovenia (0120–569/2017/4), the ethics committee of University Regensburg, Germany (18–1034–101); and the Clinical Research Ethics Committee of the Basque Country, Spain (PI2015075). All participants or their caretaker or parents gave their informed consent.

### 2.2. Study Population

At the inclusion time, saliva samples (n = 143) were collected from the SysPharmPediA participants (Figure 1). Clinical evaluation, doctor-reported medication use, inhaler device types, medication adherence, and inhaler technique assessments were conducted at the inclusion time, which was previously described in detail [14,15].

### 2.3. Saliva Sample Collection, 16S rRNA Sequencing and Processing

A detailed description of saliva sample collection, 16S rRNA isolation, sequencing, and read processing was reported previously [16]. In summary, a total of 143 saliva samples were collected at the inclusion time in falcon tubes and stored at −80 °C in each center until the shipment to a long-term storage biobank in Regensburg, Germany. Polymerase chain reaction (PCR) amplification using specific primers (hypervariable V3–V4 selective of 16S rRNA) was performed. Based on standard Illumina protocol, samples were sequenced by MiSeq V3 2 × 300 bp. Sequencing of more than 10,000 reads per sample was considered. After the quality control using MultiQC [17] and FastQC [18], primers were removed by Cutadapt [19], and then the divisive amplicon denoising algorithm-2 (DADA2) pipeline [20] was followed (GitHub: https://benjjneb.github.io/dada2/tutorial.html accessed on 29 April 2021). Furthermore, according to the SILVA database v138 [21], amplicon sequence variants (ASVs) were annotated to their respective bacterial taxonomy.

### 2.4. Statistical Analysis

Visualization techniques (Q-Q plots and histograms) and Kolmogorov–Smirnov tests were applied to assess the distribution of continuous variables. Mean ± standard deviation (SD) and median and 25th and 75th percentiles were reported for continuous variables that followed normal and non-normal distributions, respectively. Demographic and clinical characteristics of DPI and MDI device users were compared using parametric and nonparametric tests (Mann–Whitney U test, Pearson Chi-Square, or Fisher’s exact tests). According to the literature and consulting with experts in the team, confounding factors were drawn in the directed acyclic graph (DAG, Figure 2) and, if applicable, considered for adjustment in further analyses. Bacterial microbiome composition and global diversity (α- and β-diversity using phyloseq [22] and vegan [23] R-packages) were compared between DPI and MDI groups. The richness and the Shannon index were used to assess the differences in α-diversity. The Wilcoxon rank-sum test evaluated the significant difference in α-diversity. The Bray–Curtis and weighted UniFrac distance measures were used to compare the two groups’ β-diversity with *p*-values calculated by Permutational Multivariate Analysis of Variance (PERMANOVA) models after adjusting for multiple covariates (as defined in the DAG, Figure 2). A *p*-value < 0.05 was defined as significant. Differential abundance was compared between the two groups using the Analysis of Compositions of Microbiomes with the Bias Correction (ANCOM-BC) method by ANCOMBC R-package [24], including an internal normalization. Taxons present in ≥5% of samples [16,25] and covariates, as defined in the DAG (Figure 2), were included in the ANCOM-BC differential abundance model. Multiple testing was corrected using the Benjamini–Hochberg method, with 0.05 as a significant cut-off if applicable. R (version 4.2.2, 31 October 2022) and R-studio (version 2023.03.1 + 446) were used for the analyses and data visualizations.

## 3. Results

Children’s baseline demographics and clinical characteristics are shown in Table 1 (nTotal = 143, 41% females, nDPI group = 65, nMDI group = 77; one participant was excluded from the analysis due to the simultaneous usage of both device types). Children in the DPI group were significantly older than the MDI group (12.8 ± 2.2 and 10.8 ± 3, respectively; *p*-value < 0.001). There was no significant difference in sex between the DPI and MDI groups. Regarding ethnicity, a higher proportion of children were European (Caucasian) in the DPI group compared to the MDI group. There was a significant difference between the two groups in the country of inclusion. Children in the MDI group were more often included in Germany (42%) and only one participant was included from Slovenia. However, in the DPI group, patients were often recruited in Spain (42%) and a few patients (9%) in Germany. The two groups had no differences in asthma control status (controlled versus uncontrolled) and (childhood) asthma control test ((c)ACT [26,27]) scores. However, patients in the MDI group had more often severe asthma (steps 4 and 5 based on GINA guidelines [1]) compared to the DPI group (66% versus 40% had severe asthma). Children in the MDI group took higher ICS dosages and had higher daily intervals than those in the DPI group.

### 3.1. Global Diversity

According to the bioinformatics pipeline, with 100% accuracy in the mock communities, bacterial taxa were identified correctly at the genus level. In total, 2677 ASVs were identified, of which Bacteroidota, Firmicutes, and Proteobacteria were the most abundant bacterial phyla, and Prevotella, Alloprevotella, and Veillonella were the most abundant bacterial genera in the whole study population (N = 142) (Figure 3). We analyzed both alpha diversity (microbial diversity within samples) and beta diversity (microbial diversity between samples) to assess the microbial community structure. We used three indexes to measure alpha diversity: 1. Observed ASVs: The number of unique ASVs per sample. 2. Shannon index: A measure of both richness and evenness of the community. 3. CHAO1 index: An estimate of species richness and considers rare species. We found no significant differences between the groups in any of these alpha diversity measures (adjusted *p*-values: observed ASV = 0.476, Shannon index = 0.559, CHAO1 index = 0.492). Regarding the β-diversity and to compare microbial composition between samples, we used two distance measures: Bray–Curtis, which considers the abundance of ASVs, and weighted UniFrac, which incorporates both abundance and phylogenetic relationships of ASVs. We applied Permutational Multivariate Analysis of Variance (PERMANOVA) to these distance measures. PERMANOVA models showed significant differences in both Bray–Curtis (adjusted *p*-value = 0.018) and weighted UniFrac (adjusted *p*-value = 0.039) distance measures.

### 3.2. Differential Abundance

Only 450 ASVs remained after considering the ASVs presented in ≥5% of all samples. At the family level, only Carnobacteriaceae (adjusted *p*-value = 0.033), and at the genus level, both *Granulicatella* (adjusted *p*-value = 0.021) and *Aggregatibacter* (adjusted *p*-value = 0.011), showed significantly higher abundance in the MDI group compared to the DPI group (Figure 4).

## 4. Discussion

Based on our main results, the difference in β-diversity and the distinct differential abundance of certain bacterial taxa in saliva samples from children with moderate-to-severe asthma who used different inhaled corticosteroid devices (MDI versus DPI) shed light on the potential impact of these devices on the saliva microbiome. Understanding how different inhaler devices affect oral cavity microbiome composition is important, as it may have implications for personalized therapy, choosing optimal devices, and reducing adverse effects in pediatric asthma.

The lack of a significant difference in α-diversity between MDI and DPI users suggests that overall microbial richness and evenness within individual saliva samples were similar regardless of inhaler device types. However, the β-diversity analysis (microbial diversity between samples) revealed significant differences between the MDI and DPI groups. The global diversity-related findings imply that while overall α-diversity within samples may not be affected, specific compositions and taxa of the saliva bacteriome may be associated with the inhaler device type.

Identifying specific bacterial taxa with significant differential abundance between MDI and DPI users provides further insight into the potential impact of inhaler type on saliva and oral bacterial composition. The higher relative abundance of Carnobacteriaceae at the family level and *Granulicatella* and *Aggregatibacter* at the genus level in the MDI group suggests that these taxa may be associated with factors unique to MDI usage, such as formulation and administration technique. *Granulicatella* and *Aggregatibacter* members are among normal oral flora bacteria that may cause serious infections and diseases like infective endocarditis and aggressive periodontitis [28]. *Aggregatibacter* genera, previously known as *Actinobacillus* [29], is a Gram-negative bacteria from the Pasteurellaceae family and is associated with periodontal infection [30,31], endocarditis, and pneumonia [32]. A longitudinal study of 700 adolescents showed the association between aggressive periodontitis and *Aggregatibacter actinomycetemcomitans* in plaque samples [33]. In addition, a systematic review with meta-analysis reported a significant association between periodontal disease and asthma [34]. *Granulicatella* is from the Carnobacteriaceae family and is a Gram-positive lactic acid bacterium [28]. *Granulicatella* species were shown to be associated with periodontitis [35], severe childhood caries [36], and endodontic infections [37,38]. Understanding the role of these bacterial taxa in oral health, asthma pathogenesis, and their response to corticosteroid inhalation and deposition could provide valuable insights into disease management strategies.

ICS treatment has been reported as a significant contributor to the variability in oral microbiome compositions. Local deposition of ICS may lead to side effects like dysphonia and oropharyngeal candidiasis, which can be prevented by rinsing the mouth and throat with water after use [3,13,39,40]. These side effects add an extra burden to the disease burden on children and their caregivers. Increased doses of ICS intake may result in greater deposition of corticosteroids in the oral cavity and may have a higher impact on oral microbiota composition. Nevertheless, we included the daily doses of ICS intake in our analyses. We have adjusted the analyses, such as global diversity (α- and β-diversities) and differential abundance analyses (ANCOM-BC) for this confounding factor.

Recent studies have elucidated the complex association between environmental exposures and pediatric asthma. Zanobetti et al. showed that early-life exposures matter in pediatric asthma. They found that fine particulate matter (PM2.5) and nitrogen dioxide (NO_2_) were associated with elevated asthma incidence in both early and middle childhood. Notably, this association was stronger among black children and in communities with fewer opportunities and resources [41]. Focusing on children aged 4–6 with a history of bronchiolitis, Dearborn et al. observed that exposure to modest postnatal ozone (O_3_) concentrations increased the risk of asthma and wheezing [42]. These findings underscore the importance of considering both individual and community-level factors when studying pediatric asthma. Children with moderate-to-severe asthma were included in this study from four European countries (The Netherlands, Germany, Spain, and Slovenia). Different atmospheric characteristics and national clinical guidelines between these counties may influence clinical outcomes, medication patterns, and the microbiome profile of individuals. To adjust for these differences, as shown in the DAG, we added the country/center of inclusion as a possible confounding factor in the analysis.

Despite demonstrating lower lung function, children in the DPI group received lower ICS dosages with less frequent intervals and showed a comparable level of asthma control to the MDI group (62%). Additionally, the DPI group exhibited a higher reversibility response to salbutamol, indicating more room for improvement in their lung function. It is important to note that asthma severity should be taken into account, as fewer children in the DPI group had severe asthma (40%) compared to the MDI group (66%). Children with lower lung function may have less respiratory force to inhale and sufficiently transfer corticosteroids into their lungs. So, lower lung function may lead to higher corticosteroid deposition, which should be taken into account for further research.

This study had multiple strengths. It is the first study to focus on assessing the association between inhaler devices and bacterial microbiome in saliva. It is a multinational European cohort, which increases the generalizability of findings compared to single-center studies. In this study, comprehensive medication intake information, including medications, dosage, intervals, and device type from the last year before inclusion time and/or at the inclusion, was collected by physicians to obtain more accurate data and report more reliable results. In addition, we could assess medication adherence and inhaler techniques and incorporate them into the analyses, which is vital in investigating the association between inhaler devices and saliva microbiome.

There are also limitations related to this study. We do not have information about oral health that can influence the oral microbiome and bacterial composition in saliva. In addition, we have not assessed fungal information from saliva samples; therefore, further research should investigate the link between the salivary fungal and bacterial composition in relation to inhaler device usage. Moreover, although we assessed medication adherence and inhaler techniques using validated methods, more objective methods like digital inhalers are suggested for future research.

Our findings contribute to a better understanding of the association between inhaler types and oral microbiome and may help optimize personalized treatment in pediatric asthma in the future. The findings indicate that inhaler device types should be recognized as potential confounding factors in asthma microbiome studies. However, more research is needed to further validate the findings and investigate the causal relation between inhaler devices and saliva microbiome.

## 5. Conclusions

The findings of this study underscore the importance of considering inhaler type as a potential factor influencing saliva microbiome composition in children with moderate-to-severe asthma. Further research is needed to elucidate the underlying mechanisms driving these differences and to determine the clinical implications of altered microbiome composition on asthma outcomes and oral health. Our findings may help the selection of inhaler devices based on individual patient characteristics and potential impacts on the oral microbiome. Understanding the impact of inhaler devices on microbiome composition and microbial dysbiosis may ultimately inform personalized asthma management strategies tailored to individual microbial profiles. These insights may ultimately improve future treatment outcomes and quality of life for children with moderate-to-severe asthma.

## Figures and Tables

**Figure 1 biomedicines-13-00089-f001:**
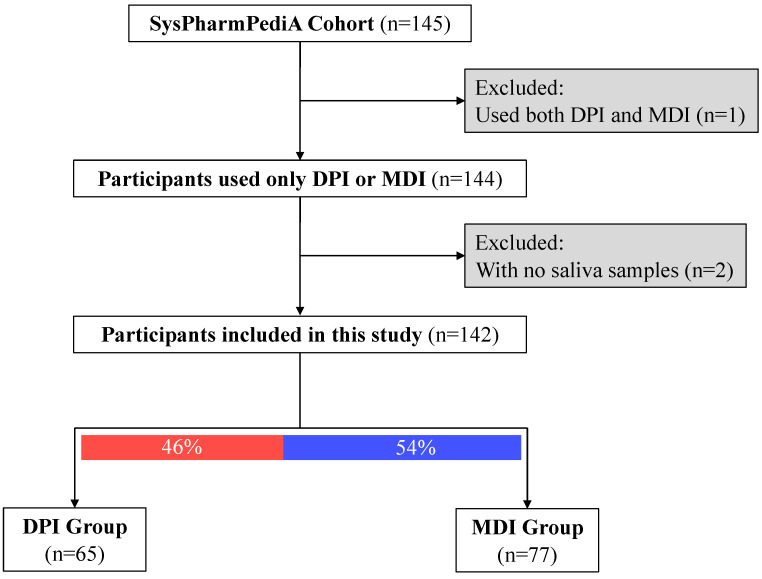
Descriptive study flow chart. Participants were categorized based on the type of inhaled corticosteroid device: DPI: dry powder inhalers; MDI: metered-dose inhalers.

**Figure 2 biomedicines-13-00089-f002:**
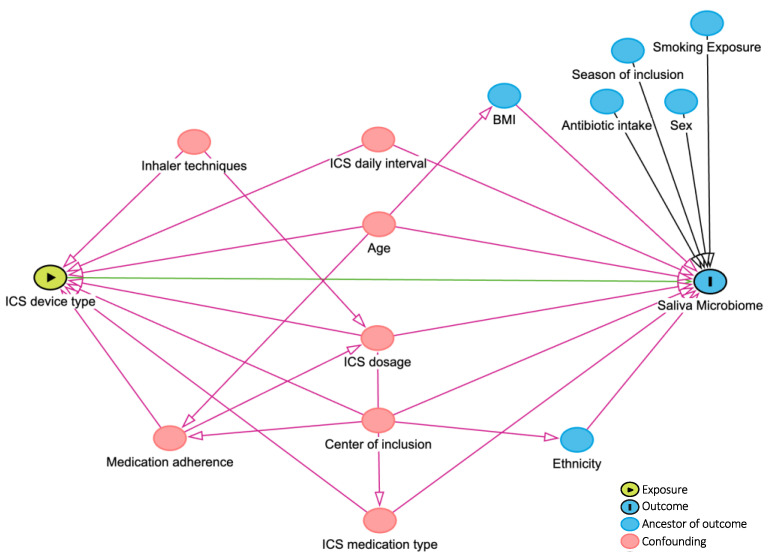
DAG (directed acyclic graph). Created by www.dagitty.net. ICS: inhaled corticosteroid.

**Figure 3 biomedicines-13-00089-f003:**
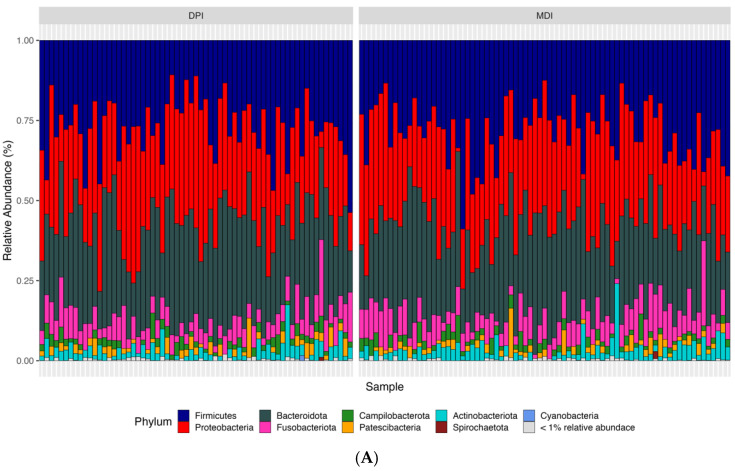
Relative abundance of samples at the phylum (**A**), family (**B**), and genus (**C**) levels between the children using DPI and children using MDI. DPI: dry powder inhalers; MDI: metered-dose inhalers.

**Figure 4 biomedicines-13-00089-f004:**
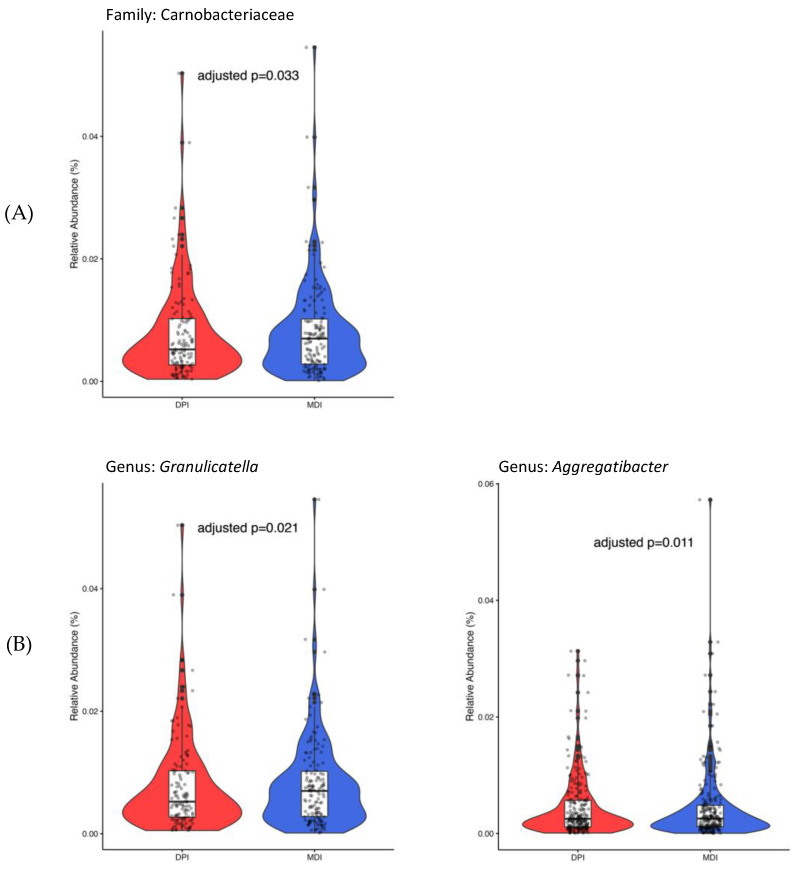
Bacterial taxa with significant differential abundance between children using DPI (in red) and children using MDI (in blue) at the family (**A**) and genus (**B**) levels. DPI: dry powder inhalers; MDI: metered-dose inhalers.

**Table 1 biomedicines-13-00089-t001:** Demographic and clinical characteristics of the participants in DPI and MDI groups.

Characteristics	DPI (n = 65)	MDI (n = 77)	*p*-Value	Total (n = 142)
Demographics				
Age, years, mean (SD)	12.8 (2.2)	10.8 (3.0)	**<0.001**	11.7 (2.9)
Female, n (%)	28/65 (43%)	30/77 (39%)	0.619	58/142 (41%)
Ethnicity, n (%)				
Caucasian	56/65 (86%)	54/77 (70%)	**0.023**	110/142 (23%)
Non-Caucasian	9/65 (14%)	23/77 (30%)		32/142 (77%)
Body mass index (BMI) z-score, mean (SD)	0.47 (1.37)	0.57 (1.21)	0.637	0.52 (1.28)
Smoking exposure, n (%)	22/61 (36%)	18/74 (24%)	0.137	40/135 (30%)
Country of inclusion, n (%)				
Spain	27/65 (42%)	23/77 (30%)	**<0.001**	50/142 (35%)
Germany	6/65 (9%)	32/77 (42%)		38/142 (27%)
The Netherlands	10/65 (15%)	21/77 (27%)		31/142 (22%)
Slovenia	22/65 (34%)	1/77 (1%)		23/142 (16%)
Clinical Characteristics				
Asthma control status, n (%)				
Uncontrolled	40/65 (62%)	48/77 (62%)	0.922	88/142 (62%)
Asthma severity, n (%)				
Moderate	39/65 (60%)	26/77 (34%)	**0.002**	65/142 (46%)
Severe	26/65 (40%)	51/77 (66%)		77/142 (54%)
ACT score, median (IQR)	23 (20, 25) (n = 65)	22 (18, 24) (n = 72)	0.401	23 (20, 25) (n = 137)
Lung function test, median (IQR)				
FEV_1_% predicted pre-salbutamol	90.7 (81.1, 98.2) (n = 64)	97.5 (85.8, 106.1) (n = 75)	**0.010**	93.7 (82.7, 103.3) (n = 139)
FEV_1_% predicted post-salbutamol	98.3 (89.0, 103.2) (n = 64)	102.6 (92.0, 110.7) (n = 73)	**0.019**	99.6 (89.6, 108.6) (n = 137)
Bronchodilator reversibility (change in FEV_1_ ≥ 12% after salbutamol intake), n (%)	21/64 (33%)	11/73 (15%)	**0.014**	32/137 (23%)
ICS type, n (%)				
Beclomethasone	4/65 (6%)	11/77 (14%)	**<0.001**	15/142 (11%)
Budesonide	14/65 (22%)	0/77 (0%)		14/142 (10%)
Ciclesonide	0/65 (0%)	1/77 (1%)		1/142 (1%)
Fluticasone	47/65 (72%)	65/77 (84%)		112/142 (79%)
ICS dosage *, n (%)				
Low	40/65 (62%)	26/77 (34%)	**<0.001**	66/142 (46%)
Medium	10/65 (15%)	36/77 (47%)		46/142 (32%)
High	15/65 (23%)	15/77 (19%)		30/142 (21%)
ICS intervals (per day), n (%)				
1	15/65 (23%)	1/77 (1%)	**<0.001**	16/142 (11%)
2	45/65 (69%)	34/77 (44%)		79/142 (56%)
3	1/65 (2%)	0/77 (0%)		1/142 (1%)
4	3/65 (5%)	40/77 (52%)		43/142 (30%)
6	1/65 (2%)	2//77 (3%)		3/142 (2%)
Spacer used with ICS device, n (%)	—	72/77 (93.5%)	—	—
Medication adherence based on MARS-5 questionnaire, n (%)				
Nonadherent (score < 23)	14/61 (23%)	15/67 (22%)	0.939	29/128 (23%)
Adherent (score ≥ 23)	47/61 (77%)	52/67 (78%)		99/128 (77%)
Inhaler technique score, median (IQR)	100 (100, 100) (n = 54)	91 (91, 100) (n = 42)	**<0.001**	100 (91, 100) (n = 96)

FEV1: Forced expiratory volume in one econd; ICS: Inhaled CorticoSteroids. * Based on GINA guideline 2016, Box 8, page 15. *p*-Values in bold indicate statistically significant differences at *p* < 0.05.

## Data Availability

For clinical and other data generated within the SysPharmPediA study, the authors will make them available upon specific requests subject to the requestor obtaining ethical, research, data access, and collaboration approvals from the SysPharmPediA study management board. Requests can be sent to a.h.maitland@amsterdamumc.nl.

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
