# Peer review of "Association of Corticosteroid Inhaler Type with Saliva Microbiome in Moderate-to-Severe Pediatric Asthma"

_biomedicines, 2025, doi:10.3390/biomedicines13010089_

Round 1

Reviewer 1 Report

Comments and Suggestions for Authors

This study is about the relationship between corticosteroid inhaler type and salivary microbiome in moderate to severe pediatric asthma. It contains very important data. The study covers many countries. I have many suggestions for the development of the study.

1- Some parts in the introduction section should be removed and some parts should be shortened.

2- Literature is missing in several places in the discussion section.

3- The discussion section should be expanded and discussions of samples taken from different countries should be added. In particular, the atmospheric characteristics of different countries have a direct effect on asthma cases. In this context, if there are any studies on the effects of atmospheric characteristics on pediatric asthma cases in countries, discussion should be made.

It is recommended that the article be published after these corrections.

Comments on the Quality of English Language

There are some grammatical errors in the article. I recommend having it checked by a native English speaker.

Author Response

We are grateful for the constructive feedback provided by the reviewer. We will carefully address each comment and make appropriate revisions to improve our manuscript's quality.

  • Introduction: We have revised the introduction section, removing unnecessary parts and shortening others to improve conciseness and focus.

A summary of changes in the introduction based on your comment:

Deleted: (second paragraph) Recent studies have begun to explore the complex link between ICS use and the airway’s microbiome composition. Some studies suggest that ICS may alter the composition and diversity of airway microbiota, though the nature and degree of these alterations are still unclear [3, 4]. Local deposition of ICS in the oropharynx and larynx may lead to side effects like dysphonia (hoarse voice), and oropharyngeal candidiasis (thrush). The severity and frequency of side effects depend on medication type, dose, administration rate, inhaler techniques, and device type, which can be prevented by rinsing the mouth and throat with water after using all ICS types [5-7]

Revised: (Third paragraph lines 83-85) The lung microbiome is influenced by the upper respiratory tract and oropharynx microbiome [11]. Saliva offers a non-invasive sampling method for microbiome studies, which is particularly beneficial for children [12]. It has been previously shown that ICS treatment is associated with airway microbiome composition and diversity [3,13]. However, more research is needed to understand the association between ICS device types and salivary microbiome in children to optimize risk assessment, monitoring, and management strategies to protect patients' oral and overall health under ICS treatments.

  • Missing references in the discussion: The first paragraph is our own results, and there are no references for this. It probably was not clear; therefore, we clarified this paragraph and made it more clear in the text.

The new version of the first paragraph of the discussion: (line 287)

Based on our main results, the difference in β diversity and the distinct differential abundance of certain bacterial taxa in saliva samples from children with moderate to severe asthma who used different inhaled corticost…

  • Discussion: We have expanded the discussion section and added recent references about the link between atmospheric characteristics and pediatric asthma.

Added section in the discussion: (lines 348-362)

Recent studies have elucidated the complex association between environmental exposures and pediatric asthma. Zanobetti et al. showed that early-life exposures matter in pediatric asthma. They found that fine particulate matter (PM2.5) and nitrogen dioxide (NO2) were associated with elevated asthma incidence in both early and middle childhood. Notably, this association was stronger among black children and in communities with fewer opportunities and resources [41; Zanobetti et al. Early-Life Exposure to Air Pollution and Childhood Asthma Cumulative Incidence in the ECHO CREW Consortium. JAMA Netw Open 2024]. Focusing on children aged 4-6 with a history of bronchiolitis, Dearborn et al. observed that exposure to modest postnatal ozone (O3) concentrations increased the risk of asthma and wheeze [42; Dearborn et al. Role of Air Pollution in the Development of Asthma Among Children with a History of Bronchiolitis in Infancy. Epidemiology 2023]. These findings underscore the importance of considering both individual and community-level factors when studying pediatric asthma. Children with moderate to severe asthma were included in this study from four European countries (the Netherlands, Germany, Spain, and Slovenia). Different atmospheric characteristics and national clinical guidelines between these counties may influence clinical outcomes, medication patterns, and the microbiome profile of individuals. To adjust for these differences, as shown in the DAG, we added the country/center of inclusion as a possible confounding factor in the analysis.

  • We acknowledge your comment regarding grammatical errors in the manuscript. We have double-checked the English throughout the manuscript and ensured that it was correct.
  • PDF Comments: We have addressed all comments in the attached PDF file, making the suggested changes to the introduction and expanding the discussion.

Reviewer 2 Report

Comments and Suggestions for Authors

This manuscript presents valuable insights into a critical topic in medicinal therapy: the use of metered dose inhalers (MDIs) and dry powder inhalers (DPIs) to deliver inhaled corticosteroids (ICS). The study highlights that the type of ICS device used is linked to variations in the saliva microbiome compositions of children with moderate-to-severe asthma. However, from a scientific perspective, this study has several limitations, including the following:

Please see the report,

Author Response

Thank you for your thorough review and valuable feedback on our manuscript. We appreciate your insights and have addressed each one of your comments as follows:

  • The introduction has been revised, and recent references for the last 2 years have been added.

Added text and references: (lines 61-64)

A recent longitudinal study on the asthma population showed that ICS use can alter airway microbiome, including both fungal and bacterial microbiota [3; Huang et al. Effect of inhaled corticosteroids on microbiome and microbial correlations in asthma over a 9-month period. Clin Transl Sci 2022]. Upper-airway microbiome (nasal and saliva) has been reported to be associated with asthma exacerbation among asthma patients on ICS therapy [4; Perez-Garcia et al. The upper-airway microbiome as a biomarker of asthma exacerbations despite inhaled corticosteroid treatment. J Allergy Clin Immunol 2023].

  • The Materials and Methods section has been reviewed again and adapted.
  • Table 1 and the result section have been reviewed, and the text has been adapted. Specifically, the country of inclusion and different environmental exposures that may influence the results have been added to the discussion.

Added section in the discussion: (lines 348-362)

Recent studies have elucidated the complex association between environmental exposures and pediatric asthma. Zanobetti et al. showed that early-life exposures matter in pediatric asthma. They found that fine particulate matter (PM2.5) and nitrogen dioxide (NO2) were associated with elevated asthma incidence in both early and middle childhood. Notably, this association was stronger among black children and in communities with fewer opportunities and resources [41; Zanobetti et al. Early-Life Exposure to Air Pollution and Childhood Asthma Cumulative Incidence in the ECHO CREW Consortium. JAMA Netw Open 2024]. Focusing on children aged 4-6 with a history of bronchiolitis, Dearborn et al. observed that exposure to modest postnatal ozone (O3) concentrations increased the risk of asthma and wheeze [42; Dearborn et al. Role of Air Pollution in the Development of Asthma Among Children with a History of Bronchiolitis in Infancy. Epidemiology 2023]. These findings underscore the importance of considering both individual and community-level factors when studying pediatric asthma. Children with moderate to severe asthma were included in this study from four European countries (the Netherlands, Germany, Spain, and Slovenia). Different atmospheric characteristics and national clinical guidelines between these counties may influence clinical outcomes, medication patterns, and the microbiome profile of individuals. To adjust for these differences, as shown in the DAG, we added the country/center of inclusion as a possible confounding factor in the analysis.

  • The global diversity section has been revised and clarified. We used the 16S rRNA method and could only study the bacteria profile. Unfortunately, with this method, we could not study fungi and viruses, which is now mentioned in the limitation of the study.

Revised section (Global diversity lines 241-268)

According to the bioinformatics pipeline, with 100% accuracy in the mock communities, bacterial taxa were identified correctly at the genus level. In total, 2677 ASVs were identified, of which Bacteroidota, Firmicutes, and Proteobacteria were the most abundant bacterial phyla, and Prevotella, Alloprevotella, and Veillonella were the most abundant bacterial genera in the whole study population (N=142) (Figures 3). We analyzed both alpha diversity (microbial diversity within samples) and beta diversity (microbial diversity between samples) to assess the microbial community structure. We used three indexes to measure alpha diversity: 1. Observed ASVs: The number of unique ASVs per sample. 2. Shannon index: A measure of both richness and evenness of the community. 3. CHAO1 index: An estimate of species richness and considers rare species. We found no significant differences between the groups in any of these alpha diversity measures (adjusted p-values: observed ASVs = 0.476, Shannon index = 0.559, CHAO1 index = 0.492). Regarding the β diversity and to compare microbial composition between samples, we used two distance measures: Bray-Curtis, which considers the abundance of ASVs, and weighted UniFrac, which incorporates both abundance and phylogenetic relationships of ASVs. We applied permutational multivariate analysis of variance (PERMANOVA) to these distance measures. PERMANOVA models showed significant differences in both Bray-Curtis (adjusted p-value=0.018) and weighted UniFrac (adjusted p-value=0.039) distance measures.

  • We have expanded the conclusion to highlight the significance of our work for future research and clinical practice. We have included potential implications for personalized asthma management and suggested directions for future studies.

The new version of conclusion: (lines 400-410)

The findings of this study underscore the importance of considering inhaler type as a potential factor influencing saliva microbiome composition in children with moderate-to-severe asthma. Further research is needed to elucidate the underlying mechanisms driving these differences and to determine the clinical implications of altered microbiome composition on asthma outcomes and oral health. Our findings may help the selection of inhaler devices based on individual patient characteristics and potential impacts on the oral microbiome. Understanding the impact of inhaler devices on microbiome composition and microbial dysbiosis may ultimately inform personalized asthma management strategies tailored to individual microbial profiles. These insights may ultimately improve future treatment outcomes and quality of life for children with moderate-to-severe asthma.

Reviewer 3 Report

Comments and Suggestions for Authors

Authors of this study aimed to compare the saliva microbiome in children with moderate-to-severe asthma using ICS via MDIs versus DPIs. For this purpose, saliva samples were collected from 143 children, aged 6-17 yrs, with moderate-to-severe asthma across four European countries and were subjected to 16S rRNA sequencing. Microbiome was compared using global diversity (α and β) between two groups of participants based on inhaler devices. They found out that no significant difference was observed in α-diversity but significant difference was found regarding β-diversity analysis  between groups. In addition, significant differential abundance between groups, with higher relative abundance in the MDI group compared to the DPI group, was detected at the family level [Carnobacteriaceae ] and at the genus level [Granulicatella and Aggregatibacter]. They concluded, accordingly, that types of ICS devices are associated with different saliva microbiome composition in moderate-to-severe pediatric asthma and  the causal relation between inhaler types and changes in saliva microbiota composition needs to be further evaluated.

This international study revealed some new findings about the above mentioned topic. It is, generally, well written and presents results in scientifically sound way. It is important topic due to the fact that asthma is very common in children. They accurately and self-critically described limitations of the study as well. However, there are some issues that should be addressed before publication process continues:

1. Many results are presented in text as well as in table 1. This is doubling of data presentation and should be avoided and presented either in text or in table but not in both of them.

2. chapter 3.1. Global Diversity

According to the bioinformatics pipeline, with 100% accuracy in the mock communities, bacterial taxa were identified correctly ............................... served unique ASV per sample (adjusted p-value= 0.476), Shannon index (adjusted p- 197 value=0.559), or CHAO1 index (adjusted p-value=0.492). Regarding the β diversity (microbial diversity between samples), PERMANOVA models showed significant differences in both Bray-Curtis (adjusted p-value=0.018) and weighted UniFrac (adjusted p- 200 value=0.039) distance measures.

- Comment: this text is very difficullt to follow and should be rewritten in order to be better understood by potential readers. In addition, abbreviations should be explained as well as presented indices even though they have been described in other articles.

3. Figures 3 and 4 present, in fact, the same findings only in a slightly different graphical way. This is doubling of data presentation as well and should be avoided and presented in a single figure.

4. Clinical implications are discussed in a very general way with a lack of more precise correlations of saliva microbiome with signs of airway inflammation (which is crucial and constant in asthma) and clinical outcomes.

Author Response

Thank you for taking the time to read our manuscript and provide us with valuable and helpful comments. We have carefully considered your comments and have made the following revisions:

  • We have removed the redundant presentation of data in both text and Table 1. The demographic and clinical characteristics are now presented only in Table 1, with brief highlights in the text. We do think it is important that the important findings can also be found in the written text.
  • Global Diversity section: We have revised this section to improve clarity and readability. We have also included explanations for abbreviations and brief descriptions of the diversity measures.

Revised section (Global diversity lines 241-268)

According to the bioinformatics pipeline, with 100% accuracy in the mock communities, bacterial taxa were identified correctly at the genus level. In total, 2677 ASVs were identified, of which Bacteroidota, Firmicutes, and Proteobacteria were the most abundant bacterial phyla, and Prevotella, Alloprevotella, and Veillonella were the most abundant bacterial genera in the whole study population (N=142) (Figures 3). We analyzed both alpha diversity (microbial diversity within samples) and beta diversity (microbial diversity between samples) to assess the microbial community structure. We used three indexes to measure alpha diversity: 1. Observed ASVs: The number of unique ASVs per sample. 2. Shannon index: A measure of both richness and evenness of the community. 3. CHAO1 index: An estimate of species richness and considers rare species. We found no significant differences between the groups in any of these alpha diversity measures (adjusted p-values: observed ASVs = 0.476, Shannon index = 0.559, CHAO1 index = 0.492). Regarding the β diversity and to compare microbial composition between samples, we used two distance measures: Bray-Curtis, which considers the abundance of ASVs, and weighted UniFrac, which incorporates both abundance and phylogenetic relationships of ASVs. We applied permutational multivariate analysis of variance (PERMANOVA) to these distance measures. PERMANOVA models showed significant differences in both Bray-Curtis (adjusted p-value=0.018) and weighted UniFrac (adjusted p-value=0.039) distance measures.

  • We have deleted Figure 4 to avoid data duplication and updated figure numbers accordingly.
  • Thank you for your comment. We agree and think that there are potential implications for clinical practice in the future, but due to the limited number of children in this study, we were careful about our statements. More research is needed to mention the clinical implications of our results. However, we expanded the discussion and conclusion sections and mentioned a bit about the future potential implications of our findings.

The new version of conclusion: (lines 400-410)

The findings of this study underscore the importance of considering inhaler type as a potential factor influencing saliva microbiome composition in children with moderate-to-severe asthma. Further research is needed to elucidate the underlying mechanisms driving these differences and to determine the clinical implications of altered microbiome composition on asthma outcomes and oral health. Our findings may help the selection of inhaler devices based on individual patient characteristics and potential impacts on the oral microbiome. Understanding the impact of inhaler devices on microbiome composition and microbial dysbiosis may ultimately inform personalized asthma management strategies tailored to individual microbial profiles. These insights may ultimately improve future treatment outcomes and quality of life for children with moderate-to-severe asthma.

We believe these revisions have addressed your concerns and improved the overall quality and clarity of our manuscript. Thank you again for your valuable input.

Round 2

Reviewer 2 Report

Comments and Suggestions for Authors

The authors made all corrections and reviewed the manuscript.